# Do Attachment Orientations Relate to Coping with Crises? Lessons from a Cross-Sectional Study during the COVID-19 Pandemic

**DOI:** 10.3390/ijerph20126177

**Published:** 2023-06-19

**Authors:** Hadas Egozi Farkash, Mooli Lahad, Limor Aharonson-Daniel

**Affiliations:** 1Faculty of Health Sciences, School of Public Health, Ben-Gurion University of the Negev, Beer-Sheva 841001, Israel; 2Department of Psychology, Tel-Hai College, Upper Galilee 1220800, Israel; 3The Community Stress Prevention Centre (CSPC), Kiryat-Shmona 1101602, Israel; 4PREPARED Center for Emergency Response Research, Ben-Gurion University of the Negev, Beer-Sheva 8410501, Israel

**Keywords:** resilience, COVID-19, mental health, stress, attachment

## Abstract

This study was designed to explore whether attachment orientations were related to distress and resilience during the COVID-19 pandemic. The sample included 2000 Israeli Jewish adults who answered an online survey during the first phase of the pandemic. The questions referred to background variables, attachment orientations, distress, and resilience. Responses were analyzed using correlation and regression analyses. A significant positive relationship was found between distress and attachment anxiety, and a significant negative relationship was found between resilience and attachment insecurities (avoidance and anxiety). Women suffered higher distress, as did people with lower income, poor health, secular religious affiliation, a lack of a sense of spacious accommodation, and a dependent family member. The findings indicate that attachment insecurities are associated with the severity of mental health symptoms during the peak period of the COVID-19 pandemic. We recommend strengthening attachment security as a protective factor for psychological distress in therapeutic and educational settings.

## 1. Introduction

The COVID-19 pandemic has adversely affected mental health worldwide. Learning to live with its impacts and sequelae requires expanding our understanding of the risk and protective factors for the distress it causes, aiming to reduce the long-term impact of mental disorders. The most common psychological reactions to the COVID-19 pandemic were anxiety, depression, loneliness, stress, and traumatic symptoms, especially during the early stages of the pandemic [1] and lockdowns [2].

Most recent studies on the mental effects of the COVID-19 pandemic investigated background and personality variables [3,4,5]. The pandemic, which caused concern and distress, may have activated the attachment system. Several studies have investigated the relationship between attachment orientations and symptoms of emotional distress during the COVID-19 pandemic [6,7,8,9] and interventions based on attachment theory [10,11,12]. The current study was designed to explore the relationship of attachment orientations to mental health during COVID-19, aiming to construct effective interventions.

### 1.1. Resilience

Resilience is successful functioning or adaptation in the face of significant adversity [13]. Lahad [14], who suggested that resilience manifests itself through a person’s coping modes, developed the BASIC Ph model, identifying different coping styles under stress. However, resilience extends beyond the individual to the community, whose resilience is defined as the community’s ability to withstand crises or disruptions [15]. Resilience was found to be vital to effective coping with the adversities of the COVID-19 pandemic ([16,17]).

### 1.2. Attachment Theory

The attachment theory was introduced by John Bowlby [18,19,20] and further developed by Mary Ainsworth [21,22]. Ainsworth demonstrates the critical impact of early relationships on personality and mental health throughout one’s lifespan and over generations. Ainsworth identified three attachment styles: secure, anxious, and avoidant. An additional category, a disorganized attachment style, was identified later by Main et al. [23].

The infant and caregiver interactions are internalized into “internal working models” that remain into adulthood [18]. These schemas of social situations provide a template for interpersonal behavior, known as the adult attachment orientation [21,24]. 

Studies of the factors that facilitate the development of attachment security in adulthood [25] revealed that people use their romantic partners as attachment figures and as a safe base [26,27]. Additionally, the transition to parenthood [28], the meaning of life events [29], and therapy [30] have been found to enhance attachment security in adulthood [31].

Adult attachment orientations play an essential role in determining the distress level of people during stressful events [25,26,28,29] through their effects on risk perception and defense mechanisms [7].

### 1.3. Attachment Orientation and Resilience

“Attachment and resilience theories have developed as two separate bodies of knowledge and the concepts regarded as complementary” [32] (p. 1). Galatzer-Levy and Bonanno [33] found that attachment orientation affects resilience, but resilience does not affect attachment orientation. The early relationship between the child and the primary caregiver determines an individual’s resilience level and the resources at their disposal to face future adversity [34,35]. 

Secure attachment is one of the critical developmental tasks of early childhood [36,37]. Evidence suggests that attachment security is associated with higher self-efficacy [38], higher self-care [39], and higher emotion regulation [40], and that those factors predict resilience prospectively [37,41,42]. Secure attachment is an internal psychological resource and a factor of resilience, fostering positive and adaptive coping with stressful life events. Conversely, attachment insecurities (avoidant, anxious, and disorganized) were found to be a risk factor for the development of adjustment difficulties, maladaptive coping behaviors, and psychiatric disorders [43,44,45,46], related to lower resilience. Attachment insecurities may increase vulnerability in coping with stress while characterized by inadequate distress regulation and blocking necessary internal resources to deal with stressful situations [47,48].

Most empirical studies on resilience and attachment have investigated attachment security as it relates to positive functioning. Studies of attachment avoidance during times of stress have yielded mixed and controversial results. Thus, Karreman [42] found no significant association between attachment avoidance and resilience, nor did Mikulincer et al. [49] indicate that attachment avoidance strategies could be inadequate in distressed situations. However, Fraley and Shaver [50] suggested that avoidant individuals have effective defenses for blocking negative thoughts and emotional arousal.

A literature review on attachment orientation and mental health burden during the COVID-19 pandemic identified only a few studies, indicating that attachment security appeared to be a protective factor for psychological disorders during COVID-19, while attachment anxiety was found to be a risk factor for distress [6,9,12,51,52]. More research is needed to expand our understanding of the relationships between attachment orientation and mental coping with the COVID-19 pandemic. Research-based information on attachment is crucial for generating applications of attachment theory in clinical practice and cultivating an internal working model in families, educational, communities, and residential environments.

The current study examined the relationship between personal and community resilience, attachment orientation, self-reported stress levels, COVID-19-related concern, loneliness, and COVID-19-related traumatic symptoms.

### 1.4. The Study Objectives Were

To examine the association between attachment orientation and psychological distress related to the COVID-19 pandemic.

To explore the association between attachment orientation and resilience factors (personal and community) during a peak period of the COVID-19 pandemic.

### 1.5. Hypothesis

Adjusted for demographic and psychological variables, people with attachment insecurities will show increased psychological distress and decreased resilience.

## 2. Materials and Methods

An online survey (6–9 April 2020) was conducted during the first phase of the pandemic. Midgam Panel, an Israeli internet-research company, distributed the questionnaire to an online panel that gets paid for its services (www.midgampanel.com, accessed on 6 April 2020). There are some 200,000 participants in Midgam’s pool, and they are sampled through stratified quota sampling. The Institutional Review Board of the Faculty of Health Sciences at the Ben-Gurion University of the Negev pre-approved the study (No. 41-2020). The survey was conducted in Hebrew, the official language spoken in Israel, and was preceded by an introduction detailing the study’s objectives. Participants were advised that completing the survey was voluntary, that they could withdraw at any time, and that their anonymity was assured (only the Midgam Panel knows the participants’ identities).

### 2.1. Participants

Of the 2302 panelists who responded to the survey, 2000 provided complete responses. On March 2023, Israel’s population was estimated at 9.73 million residents. A total of 7.145 million people were Jewish (73.6% of the total population), 2.048 million people were Arab (21%), and 534,000 people were others (5.5%) [53]. Participants were Israeli adults, all Jewish (age 18–74 years), of diverse levels of religiosity and socioeconomic backgrounds. Table 1 presents the demographic characteristics of the respondents.

### 2.2. Measures

Attachment: The ECR-12, a short version of the Experiences in Close Relationships scale [54], was used to measure attachment. The questionnaire consists of 12 items (e.g., “I feel comfortable being dependent on other people”) and measures avoidant and anxious attachment. Participants were asked to rate these items on a 7-point Likert scale (1 = strongly disagree, 7 = strongly agree). The Avoidant Attachment score was reached by averaging the six odd-numbered items, with higher scores indicating greater avoidance. The Attachment Anxiety score was reached by averaging the six even-numbered items, with higher scores indicating greater anxiety and low scores on the two dimensions reflecting Secure Attachment. The ECR-12 has been used in clinical and community samples, and its psychometric properties are reported to be similar to those of the original 36-item ECR scale [54].

The questionnaire demonstrated moderate internal reliability (α = 0.65). The avoidance attachment questionnaire showed good internal reliability (α = 0.76), and the anxious attachment questionnaire revealed good internal reliability (α = 0.79).

Traumatic symptoms related to COVID-19. This questionnaire has been constructed based on three validated questionnaires: the Clinician-Administered PTSD Scale for DSM-5 (CAPS-5) [55], the Posttraumatic Diagnostic Scale for DSM–5 (PDS) [56], and the Patient Health Questionnaire (PHQ-9) [57]. The 18 items measured factors such as fear/helplessness, rage/betrayal, and exhaustion/detachment. (e.g., “I feel distant and detached from the people around me”). Moreover, they have been designed to reveal ongoing traumatic symptoms associated with COVID-19, as per DSM-5 criteria [58]. Responses have been rated on a four-point Likert scale (0 = not at all, 3 = very much), and internal reliability was high (α = 0.90).

COVID-19-related concern. The 7-item Situational Anxiety Questionnaire [48], based on the State-Trait Anxiety Inventory (STAI) [59], was the tool for examining concerns. The statements (e.g., “I am worried about my health because of the coronavirus”) have been rated on a 5-point Likert scale (0 = not at all, 4 = very much). Internal reliability was high (α = 0.88).

Stress. Stress was measured with five items that examined anxiety, stress, nervousness, sadness, and anger, based on the Perceived Stress Scale (PSS) questionnaire developed by Cohen et al. [60]. The statements (e.g., “How anxious are you these days?” and “How angry are you these days?”) have been rated on a 5-point Likert scale ranging from 0–4 (0 = not at all, 4 = very much). Internal reliability was high (α = 0.96).

Loneliness. Loneliness was measured using three items from the 20-item revised UCLA Loneliness Scale [61]. The statements (e.g., “How many times in the last month have you felt: that you are alone”) have been rated on a 5-point Likert scale (0 = never, 4 = always). The Hebrew version was developed and validated by Hochdorf [62] and has been used in various studies in Israel. Internal reliability was good (α = 0.80).

Personal resilience. Personal resilience was measured on the 10-item Conor Davidson scale (CD-RISC10) [13], and the statements (e.g., “I can achieve my goals”) have been rated on a 5-point Likert scale (0–not at all, 4–largely correct). Internal reliability in the present study was high (α = 0.88).

Community resilience. We used a modified version of the Conjoint Community Resiliency Assessment Measure (CCRAM)-10 [15]. The six items (e.g., “I feel a sense of belonging to where I live”) were rated on a 5-point Likert scale (0 = strongly disagree, 4 = strongly agree). Internal reliability was high (α = 0.89).

Background information. Questions referred to gender, marital status, birthplace, number and condition of dependents, health status, education, housing, religiosity, income, history of quarantine, and volunteer activity. 

### 2.3. Data Analysis

The SPSS Statistics 27 package (IBM, New York, US) was used for data analysis. We computed mean and standard deviation indices and used Cronbach’s alpha for the variables to examine each component. The calculated Pearson correlation coefficients were used to examine inter-variable associations. Next, a two-step hierarchical linear regression model was built to predict COVID-19 distress (composed of COVID-19 PTS, COVID-19-related concern, stress, and loneliness) as well as personal resilience, adjusted to demographic and psychological variables. In Step 1, demographic variables were brought under control, and in Step 2, attachment orientations (anxiety and avoidance) and other variables were introduced; *p* values are reported at a significance level of *p* = 0.05 and *p* = 0.01.

## 3. Results

The study’s objective was to examine the associations between attachment orientation and psychological distress related to COVID-19 and with resilience factors (personal and community) during the COVID-19 pandemic.

The psychological distress variables were COVID-19-related traumatic symptoms, COVID-19-related concern, loneliness, and stress.

Table 2 presents Pearson correlations between the main study variables. Supporting the study hypothesis, attachment anxiety is positively associated with COVID-19-related traumatic symptoms (r = 0.53, *p* < 0.01), stress (r = 0.51, *p* < 0.01), loneliness (r = 0.49, *p* < 0.01), and COVID-19-related concern (r = 0.48, *p* < 0.01). 

High attachment anxiety is negatively associated with personal resilience (r = −0.31, *p* < 0.01) and community resilience (r = −0.10, *p* < 0.01). 

High attachment avoidance is negatively associated with personal resilience (r = −0.14, *p* < 0.01) and community resilience (r = −0.17, *p* < 0.01).

Attachment anxiety and attachment avoidance, which reflect attachment security when low, are negatively associated with personal resilience (r = −0.31, *p* < 0.01), (r = −0.14, *p* < 0.01) respectively, and with community resilience (r = −0.10, *p* < 0.01), (r = −0.17, *p* < 0.01), respectively.

Also, stress is positively associated with COVID-19-related traumatic symptoms (r = 0.69, *p* < 0.01), with COVID-19-related concern (r = 0.66, *p* < 0.01), and with loneliness (r = 0.51, *p* < 0.01), and is negatively associated with personal (r = −0.31, *p* < 0.01) and community resilience (r = −0.13, *p* < 0.01).

### Multivariate Models

We performed two multi-linear regression models to predict: (1) COVID-19 distress (composed of COVID-19-related traumatic symptoms, COVID-19-related concern, stress, and loneliness) and (2) personal resilience. 

Predicting distress, results showed that having a dependent family member (β = 0.14, *p* < 0.05) or having a dependent person with special needs (β = 0.07, *p* < 0.05) is associated with high COVID-19-related distress. Living in a house (β = −0.09, *p* < 0.01), having a sense of space (β = −0.06, *p* < 0.05), and being religious (β = 0.08, *p* < 0.05) were associated with lower distress levels. Additionally, high income (β = −0.07, *p* < 0.05) and good self-reported health status (β = −0.12, *p* < 0.01) were also negatively associated with distress. Women reported higher levels of distress than men (β = 0.15, *p* < 0.01).

As seen in Table 3, after adjusting for demographic variables, high attachment anxiety was strongly associated with high distress (β = 0.57, *p* < 0.01). A very weak correlation was found between attachment avoidance and distress (β = 0.04, *p* < 0.05). 

Predicting personal resilience, results indicated that high income (β = 0.07, *p* < 0.01) and good health (β = 0.07, *p* < 0.01) were positively associated with resilience.

After adjusting for demographic variables, high attachment anxiety was associated with low resilience (β = −0.33, *p* < 0.01). Attachment avoidance, too, was associated with low resilience (β = −0.18, *p* < 0.01), as seen in Table 4.

## 4. Discussion

This study aimed to increase our understanding of who may be at particular risk for mental health problems and who may show higher resilience when coping with the COVID-19 pandemic, using attachment orientation and resilience as protective and risk factors. Our study supports the importance of applying attachment theory [18,21] during the COVID-19 pandemic, emphasizing the contribution of early internalized relationships to the individual’s coping skills with distress in adulthood [47,63,64]. We examined the associations between attachment orientations, psychological distress, and resilience factors among Israeli Jewish adults during the first wave of the COVID-19 pandemic, when Israel was in lockdown and before vaccines were available.

Our findings revealed associations between attachment anxiety and PTSD symptoms, psychological distress, and lower resilience during the peak period of the pandemic. The COVID-19 pandemic can be experienced as traumatic [65], possibly disrupting one’s psychological stability and causing emotional problems [66,67]. Individuals with attachment anxiety seem more vulnerable to the psychological challenges of the COVID-19 pandemic [7,12,50]—their sense of self-worth is fragile, they perceive others as insensitive, and the environment as unpredictable [68]. They lack feelings of security and are more vulnerable to death-related concerns [69]. In situations like lockdown or isolation, when they have no one to lean on, they may find it difficult to calm themselves. Instead, they might have hyperactivating strategies of affect regulation, which can lead to exaggerated threat assessment [47].

The current study revealed that individuals with an attachment avoidance orientation demonstrated a complex coping pattern with the psychological effects of the first wave of COVID-19. On the one hand, this orientation was associated with higher levels of loneliness and lower resilience during the first wave of the COVID-19 pandemic. These results are in line with previous studies that showed that avoidant defenses might be fragile, tend to break down under stress [45], and “may increase vulnerability to a broad range of illnesses” [70] (p. 237).

On the other hand, we found lower levels of psychological distress (COVID-19-related concern, stress, and COVID-19 PTS symptoms) among individuals with an attachment avoidance orientation. These findings are consistent with previous findings associating attachment avoidance with decreased distress during the COVID-19 pandemic [7,52], perhaps because these individuals are characterized by deactivating strategies that involve suppressing painful thoughts and denial of fears and worries [47]. In addition, they feel uncomfortable in social interactions and, therefore, may perceive pandemic isolation as less stressful than those with attachment anxiety. Furthermore, because avoidant people have a low need for other people, they may experience isolation and quarantine as times when they feel good and resilient and not as deviants from the rest of the population. However, Ostacoli et al. [71] concluded that avoidant individuals’ seemingly calm appearance might conceal a turbulent inner experience. Avoidant people tend to maintain physical and psychological distance to avoid the intrusion of outside threats [72] or divert their attention to positive and relaxing things that could also reduce stress [73]. We suggest that for people with an attachment avoidance orientation, isolation may be an environment that contains their coping style. The pandemic-mandated social distancing normalized their distress as it became “normal” during COVID-19. For people with attachment avoidance orientation, the isolation of COVID-19 was an unusual situation where they suddenly were not the odd ones out. This reduced their levels of distress while maintaining their core experience of loneliness and retaining their low-level resilience.

Attachment security, perceived as a personal resource mitigating adversity and promoting adaptive coping [74,75], seems to be a protective factor against the adverse psychological effects of the pandemic. Security-based strategies are characterized by active, constructive, and creative coping with negative affect [47]. This is consistent with Steele’s [49] claim that attachment security moderates the fears activated by the COVID-19 pandemic, acting “as a protective shield against the formation of emotional problems, including PTSD, following trauma” [76] (p. 10).

Our literature review yielded only one study on the association between community resilience and attachment orientation during the COVID-19 pandemic [11]. In the present study, we found a significant negative association between attachment insecurities (avoidance and anxiety) and community resilience, meaning that insecure individuals reported lower community resilience. We assume that insecure individuals have difficulties developing and maintaining community relationships during routine [11] or using community services and activities. During days of calm, this lack may be tolerated by them, but in the situation created throughout the COVID-19 pandemic, the lack of community resources and skills that were necessary to cope was intensified. Moreover, due to their lack of previous experience, it was difficult for them to use these services during the crisis. In addition, the emotional and social difficulties of individuals with attachment insecurities may have intensified due to isolation, lockdowns, and a lack of access to social support channels, all of which increased during the COVID-19 pandemic [77,78]. Thus, they have been left with reduced community resilience resources in times of emergency and crisis, highlighting that resilience is not only a resistance to threatening situations. Rather, it is the active and productive participation of the individual in social relations that is important to cultivate during routine. Resilience is associated with trust, compassion, social activities, good relations with others, and seeking support. Attachment theory extends beyond the individual and is also a lens through which broader social phenomena can be examined.

Our study also revealed personal differences in the adverse psychological effects of the pandemic. Higher levels of mental distress were found among women, people with lower income and poor health, and those with a dependent family member or a dependent person with special needs. Living in a house (rather than an apartment), having a sense of space, or being religious decreased distress levels.

In line with previous studies [79,80,81], our findings also indicated that women are at higher risk for distress during the COVID-19 pandemic, perhaps because their ruminative emotional coping style [82] is greater than that of men. Additionally, especially during the COVID-19 lockdowns, women carried a significant burden with greater care responsibilities outside and inside the home, which may have raised their stress levels. However, these findings reflect the first wave of the pandemic, and future studies should examine the pandemic’s impact on women over time.

Our findings indicated that low socioeconomic status is another risk factor for higher mental distress. These results are consistent with previous studies [83,84,85], which revealed higher psychological symptoms among individuals of low socioeconomic status, job loss, or unemployment following the COVID-19 pandemic. Lower socioeconomic status is also associated with smaller living quarters, which, in the case of COVID-19, added to stress and distress during long periods of government-imposed general quarantine. As occupational security is critical for psychological well-being [86,87], people who lost their financial stability may have also lost part of their self-identity, self-esteem, self-confidence, and meaning in life, in addition to suffering from concern about the future.

A sense of space at home was associated with less distress. During the COVID-19 pandemic, lockdowns and social distancing mandated that people spend more time at home due to governmental orders, work from home, and closures of public spaces. Social distancing, quarantine, lockdown, or isolation all have potentially harmful effects that may lead to psychological distress [88]. While for some, the home was protective, a source of space, and an anchor of security, for others, it was a risk factor for developing frustration and distress [89,90].

Poor health and having a dependent family member or a dependent person with special needs were found to be other risk factors for distress. Previous studies have shown that people with medical problems are at increased risk for adverse psychosocial outcomes of the pandemic [91,92]. Furthermore, preexisting susceptibility within families may cause unique challenges and increase the potential for mental disorders during the pandemic [93,94].

Religious belief has been positively associated with decreased distress levels. Faith can help people survive the uncertainty of COVID-19 by creating positive emotions, providing mental relaxation, and providing strength and comfort [95,96,97]. Faith gives meaning to life, which impacts mental wellbeing [98,99].

The COVID-19 prohibitions disrupted our ability to maintain close physical relationships and, at the same time, highlighted the essential role of these relationships for mental wellbeing. Our study emphasizes the importance of developing attachment security for regulating concerns and distress. However, “Although attachment styles are open to revision in adulthood, it may be relatively challenging to modify them” [25] (p. 409). Fortunately, developing a secure attachment and resilience as a result can be promoted by providing a secure base through therapy [100], mindfulness [101], or intimate relationships [12,35,102]. Additionally, parenting, significant life events such as changing jobs or starting new relationships, and the positive interpretations of such experiences [25,103] enhance a secure attachment.

Attachment theory expands the concept of resilience, assuming that relationships are key to resilience, serving as a source of trust, self-confidence, and strength [32,104]. Attachment security is associated with greater resilience and more effective coping mechanisms and strategies in times of adversity. It is connected, on the one hand, with the ability to trust oneself and with more self-efficacy and self-care [37], and on the other hand, with the ability to trust others and turn to them when needed. Thus, as it is known, resilience is associated with more flexibility, which allows the use of these two strategies of turning to others and relying on oneself as needed. According to the attachment theory, resilience is “the ability to turn to others for help and guidance when in doubt” [75] (p. 98), an ability many sought—and continue to seek—during the ongoing pandemic.

### Limitations

This study was conducted online, so those without internet access were automatically excluded from the study. However, it should be noted that, per Digital Israel [105], most Israelis (84% in January 2020 and 88% in January 2021) are connected to the internet. Another possible limitation is that the reliability of self-report-based questionnaires may be partially biased when exploring psychological concepts such as resilience, distress, and attachment. However, conducting surveys online was the best way to collect data during a lockdown.

In addition, the ECR-12 questionnaire is based on a clinical perspective that assumes absence or low negative/pathological measures to reflect strength/resilience and health. It is a known approach that the ECR 12 is based on and known to be a good measure of attachment orientation. However, we believe that future studies should use a direct measure of security attachment.

## 5. Conclusions and Recommendations

The current study illuminated the importance of attachment security as a protective factor and an element of resilience during the COVID-19 pandemic.

Fostering resilience is critical to improving support for vulnerable populations facing the psychological damage caused by the pandemic. Among these adults are individuals of low socioeconomic status, poor health, or those with special needs. Gender differences were found to be important, so there is a need to raise awareness of such differences and encourage assistance for mothers who combine home and work. Interventions promoting attachment security should be provided, considering a holistic approach that combines personality characteristics and internalized object relations.

Given that secure attachment provides emotional protection, it is worth thinking about ways to encourage people to develop it during adulthood. Findings suggest that resilience and attachment may be appropriate interventions. Attachment-based psychotherapy focuses on the bond between the therapist and the patient and recognizes the patient’s attachment dynamics, such as behavioral and emotional patterns of insecurities and psychological symptoms [30,106]. The psychotherapy goal is to restore attachment security and foster affect regulation strategies [47]. Additionally, introducing the issue into the education system as part of life skills and developing a sense of ability, security, calm, and resilience can be a task for the future.

Assuming that parents may be more anxious than their children during the COVID-19 pandemic, attachment-based interventions for providing parents with tools for self-regulation, responsiveness, and support for their child should be considered in the community [31]. Mass media and information centers for the public will do well to emphasize skills and tools available to parents, teachers, and caregivers to deepen children’s and adolescents’ sense of security. In contrast, government media agents, who mostly spread intimidation and threats, should formulate messages that support empowerment and civic capabilities.

## Figures and Tables

**Table 1 ijerph-20-06177-t001:** Demographic characteristics of respondents (*N* = 2000).

Variables	*N*	%
Gender		
Women	1004	50.2
Men	995	49.75
Marital status		
Married/cohabiting	1055	59.05
Single	915	40.95
Age		
15–24	305	15.25
25–34	436	21.8
35–44	397	19.85
45–54	320	16
55–64	309	15.45
65+	233	11.65
Children under 16		
0	884	44.2
1	315	15.7
2	227	11.3
3	130	6.5
4+	69	3.3
Number of people at home		
0	86	4.3
1	272	13.6
2	478	23.9
3	343	17.1
4	330	16.5
5	286	14.3
6+	203	10
Dependents		
Special needs	72	3.6
Dependent child	519	25.95
Dependent adult	197	9.85
Over 70	175	8.75
House Type		
Apartment	1393	69.65
House	572	28.6
Garden/view in the house	1604	80.2
Education		
University/college	885	44.25
Post-secondary education	564	28.2
High school	494	24.7
Elementary school	21	1.05
Religiosity		
Religious	793	29.16
Secular	1207	60.35
Birthplace		
Israel	1613	80.65
Not Israel	387	19.35
Health Status		
Healthy	1489	74.45
Chronic illness	477	23.85
Income		
Above average	296	14.8
Average	544	27.2
Below average	944	47.2
Quarantined	299	14.95
SARS-CoV-2 positive	8	0.4
Volunteering in a routine	471	23.5
Volunteering during COVID-19	309	15.4

**Table 2 ijerph-20-06177-t002:** Means, standard deviations, and Pearson correlations between the main research variables.

	M	SD	1	2	3	4	5	6	7
Attachment avoidance	3.45	1.19							
2. Attachment anxiety	3.14	1.29	−0.12 *						
3. COVID-19-related concern	1.95	0.91	−0.08	0.48 **					
4. Stress	1.50	0.95	−0.04	0.51 **	0.66 **				
5. COVID-19-related traumatic symptoms	0.73	0.54	0.04	0.53 **	0.47 **	0.69 **			
6. Loneliness	1.37	0.97	0.09 **	0.49 **	0.29 **	0.49 **	0.56 **		
7. Personal resilience	3.09	0.67	−0.14 **	−0.31 **	−0.17 **	−0.31 **	−0.44 **	−0.28 **	
8. Community resilience	2.45	0.95	−0.17 **	−0.10 **	−0.07	−0.16 **	−0.22 **	−0.21 **	0.23 **

* *p* < 0.05, ** *p* < 0.01.

**Table 3 ijerph-20-06177-t003:** Hierarchical regression coefficients predicting COVID-19 distress according to demographic and psychological variables (*N* = 1428).

	Model 1			Model 2		
Variable	*B*	(β)	S.E.B.	*B*	(β)	S.E.B.
Step 1						
Age	0	−0.04	0	0	0.01	0
Number of household	−0.01	−0.02	0.02	−0.03	−0.05	0.02
members under 16						
Dependent family member	0.14 *	0.08 *	0.06	0.13 *	0.08 *	0.05
A dependent person with	0.31 *	0.07 *	0.11	0.22 *	0.05 *	0.09
special needs						
Housing type	−0.16 **	−0.09 **	0.05	−0.12 **	−0.07 **	0.04
Space: Garden/view	−0.14 *	−0.06 *	0.06	−0.11 *	−0.05 *	0.05
Religiosity	−0.07 **	−0.08 **	0.03	−0.05 **	−0.06 **	0.02
Living alone	0.01	0	0.09	0.04	0.02	0.07
Marital status	−0.03	−0.02	0.09	−0.01	0	0.07
Income	−0.05 *	−0.07 *	0.02	−0.04 *	−0.06 *	0.02
Gender (female)	0.24 **	0.15 **	0.04	0.17 **	0.10 **	0.04
Number of children	−0.04	−0.08	0.02	0.01	0.02	0.02
Health	−0.24 **	−0.12 **	0.05	−0.17 **	−0.09 **	0.04
Step 2						
Attachment avoidance				0.03 *	0.04 *	0.02
Attachment anxiety				0.38**	0.57 **	0.01
*R* ^2^		0.09 **			0.38 **	
Δ*P*^2^		0.09 **			0.29 **	
*F*		10.57 **			59.27 **	

* *p* < 0.05, ** *p* < 0.01.

**Table 4 ijerph-20-06177-t004:** Hierarchical regression coefficients predicting personal resilience according to demographic and psychological variables (*N* = 1428).

	Model 1			Model 2		
Variable	*B*	(β)	S.E.B.	*B*	(β)	S.E.B.
Step 1						
Age	0	0.09	0	0	0.06	0
Number of household	−0.01	−0.01	0.02	0	0	0.02
members under 16						
Dependent family member	0.01	0.01	0.05	0.01	0	0.05
A dependent person with	−0.15	−0.04	0.09	−0.12	−0.03	0.09
special needs						
Housing type: private house	0.02	0.02	0.04	0.01	0.01	0.04
Garden/view in the house	0.1	0.05	0.05	0.07	0.04	0.04
Religiosity	−0.01	−0.01	0.02	−0.02	−0.02	0.02
Living alone	−0.07	−0.05	0.07	−0.12	−0.08	0.07
Marital status	0.02	0.01	0.07	0.02	0.01	0.07
Income	0.05 **	0.08 **	0.02	0.04 **	0.07 **	0.02
Gender (female)	−0.02	−0.02	0.03	−0.03	−0.02	0.03
Number of children	0.02	0.06	0.02	0	0.01	0.02
Health (yes)	0.11 *	0.07 *	0.04	0.07	0.05	0.04
Step 2						
Attachment avoidance				−0.10 **	−0.18 **	0.01
Attachment anxiety				−0.17 **	−0.33 **	0.01
*R* ^2^		0.03 **			0.15 **	
Δ*P*^2^		0.03 **			0.12 **	
*F*		3.62 **			16.59 **	

* *p* < 0.05, ** *p* < 0.01.

## Data Availability

The data supporting these study’s findings are available from the corresponding author upon reasonable request.

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
