# Peer review of "Do Attachment Orientations Relate to Coping with Crises? Lessons from a Cross-Sectional Study during the COVID-19 Pandemic"

_ijerph, 2023, doi:10.3390/ijerph20126177_

Round 1

Reviewer 1 Report (Previous Reviewer 1)

I accept this research in present form

Author Response

Thank you for the review.

Reviewer 2 Report (Previous Reviewer 2)

After the corrections made by the authors, I believe that the manuscript is suitable for publication.

Please be careful with the line spacing in table 1, row 3, so that the cell sizes are equal.

Mind the spaces before/after “/” on line 391

Author Response

Point 1: Please be careful with the line spacing in table 1, row 3, so that the cell sizes are equal.

Response 1: Thank you for your comment. The spacing in table 1 row 3 has been adjusted to have equal cell sizes

Point 2: Mind the spaces before/after “/” on line 391

Response 2: Thank you for your comment. The spaces before/after “/” on line 391 have been removed.

This manuscript is a resubmission of an earlier submission. The following is a list of the peer review reports and author responses from that submission.

Round 1

Reviewer 1 Report

The article analyzes very important research issues regarding the relationship between personal and community resilience, attachment style, self-reported stress levels, COVID-19-related concern, loneliness, and COVID-19-related traumatic symptoms. Attachment style was found to be a predictor of the degree of mental health burden during the COVID-19 pandemic. Secure attachment style appeared to be a protective factor for psychological disorders during COVID-19, while anxious attachment style was found to be a risk factor. This study should be pursued.

I would only have one remark that concerns the participants of the survey: “Of the 2302 panelists who responded to the survey, 2000 provided complete responses. Participants were Israeli adults, all Jewish (age 18-74 years), of diverse religious and socioeconomic backgrounds” (lines 119-120).

This sentence may be partly incomprehensible in some countries of the world. Why? Because Israel's specific demographic situation is not always remembered. On 31 December, 2022, Israel's population was estimated at 9,656,000 residents. 7,106,000 were Jews (73.6% of the total population), 2,037,000 - Arabs (21.1%) and 513,000 Others (5.3%).

I think it is worth adding this basic information about the people living in Israel. Then, it is necessary to specify precisely who the participants of the survey were.

Reviewer 2 Report

The manuscript has many weaknesses in conceptual and reporting issues as well as in data management.

The second paragraph of the introduction discusses the studies done so far on mental health and COVID-19, but the references stop at 2020. In the last two years (since 2021) numerous studies have been published on the mental health effects of attachment-related variables in times of pandemic, which should be reflected in the introduction.

I am confused about the literature. For example, the paper by Ramsauer et al., which is cited regarding how little attention has been paid to attachment-related variables in pandemic related interventions, was published in November 2019, and does not echo any other pandemic prior to 2020. It is a study on postpartum depression. It may be an error, in which case it should be corrected along with the updated review of papers addressing this topic.

I'm sorry I'm not a native speaker, I can't evaluate it accurately, but it seems to me that sometimes the language is not adequate or it is not saying what it should. On page 2, lines 79 80, it says that: “Avoidant and anxious attachment contain adaptive mechanisms with a degree of resilience that allows one to have relationships and manage emotions”. On the one hand, all of us, regardless of our attachment history, have relationships and manage emotions, some better and some worse. On the other hand, the concepts of attachment and resilience seem to be mixed in the same paragraph without it being clear what each one refers to. Third, the first part of this paragraph talks about the role of insecure attachment in future psychopathology, without this having anything to do with the issue addressed in the section: the relationship between attachment style and resilience. Finally, I think to undestand the statment; "secure attachment, is one of the critical developmental tasks of the early period", but I do not find its relation to the second half of the sentence nor to resilience. The works cited here do talk about the relationship between attachment and resilience, but I am afraid that the authors' thesis is not well reflected in the wording of the text.

The justification of the study is based on a single paper with data from Italy, from which it is considered that "lack of research-based information on attachment may generate incorrect applications of attachment theory in clinical practice". It would be necessary to explain in detail what are these possible misapplications and what is the basis for this fear.

I am also surprised by the creation of ad hoc scales based on the data obtained in the ECR-12 questionnaire. If the authors of the ECR-12 recommend this, it should be stated in the manuscript. The low alpha value is a problem that should be clearly stated in the limitations of the paper, and not as something that "may be partially" a bias.

If the median is used to create the four groups for statistical analysis, it would be important to say how many participants finally make up each group.

It is also strange that the four groups are called by the names generally attributed to infantile attachment styles (and not, since this is a study of adults, adult attachment styles). In any case, in the general population the disorganized (or fearful) attachment style is fortunately low with respect to other insecure styles; it would be particularly important to know how many participants in the study are in this group.

The self-constructed "Traumatic symptoms questionnaire" needs to be well described. Besides the example given, which seems to look for dissociative symptoms, what other traumatic symptoms does it look for?

It is not recognized in the limitations that most of the instruments used are composed of single items taken from larger instruments. Although the internal consistency appears in all cases to be good, it is important to point this out.

It should be clarified on what data results are being provided in Table 2. For example, it is not clear what is called "stress" (variable #4). It is essential to know which questionnaire it comes from and which items measure it.

Table 4 should be set up for easy interpretation. The minus signs should be on the same line as the numerals. It could be partly a problem of the configuration of the pdf file, but it is attributable to the authors that different line spacing, margins and alignments are mixed in the table and should be scrupulously corrected. The same applies to Table 3.

The paragraph from lines 195 to 204 repeats the data just presented in the table and is mostly unnecessary.

Table 1 refers to the variable "high in anxiety and avoidance", although in the general description of the measures it was called "disorganized attachment style". The authors should use a homogeneous terminology throughout the manuscript.

In the multivariate analyses, lines 233 234, the variable "COVID-19 distress" is described. It is not clear, but it could be the variable "stress" that appears in Table 2. Variables should be clearly described, in the appropriate place, and justified.

I think the wording should be much more careful.

For example, at the beginning of the discussion (lines 260-262) it is said that the present study "supports the attachment theory". The attachment theory is out of the question. What this paper supports, if anything, is the importance of applying that theory in a given context or situation.

The authors should not forget that they have done a cross-sectional study at a point in time in April 2020, and avoid statements that lead one to think that the data are representative of what happens "during the COVID-19 pandemic" (e.g., line 292).

The relationships between the construct "resilience" and the construct "attachment" are not addressed in sufficient depth. There is undoubtedly a strong relationship between them that has been studied by numerous authors and that does not appear in the discussion, as reflected by the absence of bibliographical references in the paragraph dedicated to this topic.

The authors should say in which language the study was conducted, since there is no mention of translations of the instruments into Hebrew, the official language of Israel, along with Arabic. If it was done in English, it would be necessary to explain how much of the Jewish Israeli population and in which social strata is able to cope with filling out questionnaires in English.

In general, the article is poorly written:

In line 49 a space should be added

For some reason the font size is smaller when writing some bibliographic citations. The font size is changing in some sections, such as at the bottom of page 11.

A space must be left between the table note and the following paragraph and between the tables or figures and the text.

Table titles do not need to be capitalized.

Tables and figures should follow APA guidelines regarding where to place the number and title.
